# Self-Supporting Conductive Polyaniline–Sodium Alginate–Graphene Oxide/Carbon Brush Hydrogel as Anode Material for Enhanced Energy in Microbial Fuel Cells

**Yuyang Wang \*, Huan Yang, Jing Wang, Jing Dong and Ying Duan**

School of Light Industry, Harbin University of Commerce, Harbin 150028, China
\* Correspondence: wangyuyanglover@163.com; Tel./Fax: +86-451-8486-5185

**Abstract:** Microbial fuel cells (MFCs) have exhibited potential in energy recovery from waste. In this study, an MFC reactor with a polyaniline–sodium alginate–graphene oxide (PANI–SA–GO)/carbon brush (CB) hydrogel anode achieved maximum power density with 4970 mW/m$^3$ and produced a corresponding current density of 4.66 A/m$^2$, which was 2.72 times larger than the MFC equipped with a carbon felt film (CF) anode (1825 mW/m$^3$). Scanning electron microscopy indicated that the PANI-SA-GO/CB composite anode had a three-dimensional macroporous structure. This structure had a large specific surface area, providing more sites for microbial growth and attachment. When the charging-discharging time was set from 60 min to 90 min, the stored charge of the PANI-SA-GO/CB hydrogel anode (6378.41 C/m$^2$) was 15.08 times higher than that of the CF (423.05 C/m$^2$). Thus, the mismatch between power supply and electricity consumption was addressed. This study provided a simple and environment-friendly modification method and allowed the prepared PANI–SA–GO/CB hydrogel anode to markedly promote the energy storage and output performance of the MFC.

**Keywords:** energy storage; anode modifier; composite material; green energy

## 1. Introduction

The microbial fuel cell (MFC) is an ideal device for converting chemical energy stored in organic matter directly into electrical energy using microorganisms as anode catalysts [1–5]. MFCs do not only use the biocatalyst activity of microorganisms to convert chemical bond energy stored in several organic fuel sources to generate electric current, but they also perform biodegradation and wastewater treatment. However, traditional MFCs have relatively low output current, and the charge generated by microorganisms is instantly consumed, lacking the capacity for energy storage. MFCs generate electricity without interruption in the continuous treatment of wastewater, but the generated electricity may not require immediate consumption. To address this problem, storing unused electricity can be an effective technique. Supercapacitors are also often used between power supplies and conventional capacitors because they can produce and store electrical energy within a short time [6,7]. Supercapacitor materials exhibit fast charge and discharge, which can be used in the MFC anode. A new bioanode with capacitive characteristics can replace the traditional bioanode. The new capacitive bioanode can store charge generated by electrogenic bacteria and quickly output two parts of the electricity (stored charge and generated charge) needed to release electricity, immediately achieving an increase in current output.

Previous studies have also demonstrated that the electricity production of MFCs equipped with capacitive materials (such as carbon material and polymers) as modified anodes leads to high performance [8,9]. Some researchers [10] prepared polyaniline-modified bioanodes to enhance the performance of the MFCs. In the charging-discharging experiment with a 20 min charge time and a 20 min discharge time, the results showed that the PANI-modified anode obtained a cumulative charge of 13,930.1 C/m$^2$, which was markedly

larger than the MFC with a bare carbon felt film anode (5908.1 C/m$^2$). Lai et al. [11] synthesized a conductive polymer-modified carbon cloth film anode. Additioanlly, an MFC reactor with a conductive polymer-modified bioanode obtained a maximum power density of 5.16 W/m$^3$, which was 2.66 times higher than that of the MFC with an unmodified anode. The charge transfer of the anode was promoted by PANI modification. Wang et al. [12] synthesized a PANI-modified anode in an MFC; the maximum power density and internal resistance of the MFC with a PANI-modified anode were 4 W/m$^3$ and 156 Ω, respectively, and those of the MFC with an unmodified anode were 1.7 W/m$^3$ and 358 Ω. Hou et al. [13] prepared a PANI-modified stainless steel fiber felt anode. Electrochemical impedance spectroscopy tests demonstrated that composite anodes can markedly decrease the internal resistance of MFCs. All results indicated that PANI modification of the anode was an efficient approach to improving the performance of MFCs. However, modification of the anode by PANI only was insufficient, this modification largely limits the output current of MFCs. Sodium alginate (SA) has recently drawn increasingattention because of its good biocompatibility and film-forming properties [14,15]. Li et al. [16] fabricated a porous and mat-like polyaniline/sodium alginate (PANI/SA) composite with uniform diameters at 50–100 nm and excellent specific capacitance. Wang et al. [17] constructed an MFC with a PANI–SA/NCNT/S composite-material-modified bioanode. The output power and energy storage performance of an MFC with a modified bioanode was significantly improved. A previous report [18] has shown that the combination of SA and PANI can effectively enhance the performance of MFCs. However, PANI has certain disadvantages, such as swelling, shrinkage, poor stability, and sluggish electron transfer properties, which limit the durability of MFCs [19]. This problem has been addressed by combining conducting polymers with graphene oxide (GO). The results indicated that the complex formation of conducting polymers with GO could overcome the negative effects of conducting polymers. Specifically, conducting polymers promote the electrochemical capacitance of GO and graphene materials [20–25]. Polypyrrole (PPy)/GO composites on the graphite felt (GF) electrode were successfully prepared by Lv et al. The MFC equipped with a PPy/GO-modified GF anode achieved a maximum power density of 1326 mW/m$^2$, which was significantly higher than that associated with an unmodified GF anode (166 mW/m$^2$) [26]. These results demonstrated that the PPy/GO composites were effective anode-modifying materials for improving the electricity generation and long-term stability of MFCs. Yong et al. [27] prepared a novel 3D graphene/PANI structure as the MFC anode. The results show that owing to its higher bacterial biofilm loading and higher EET efficiency, the performance of the modified anode is obviously better than that of the common carbon cloth film.

On this basis, we prepared the PANI-SA-GO composite material, which has the advantages of different materials and makes up for the shortcomings of a single material. However, similar to carbon paper and carbon felt film, and so on, the traditional substrate has a small specific surface area. Furthermore, it cannot provide the advantages of the PANI–SA–GO composite.

Natural polymer hydrogels have been widely used in the medical field because of their good biocompatibility and biodegradability. Polymer hydrogels are distinct materials in synthetic hydrogels, which exhibit the dual properties of hydrogels and organic conductors. Owing to their inherently porous structure and high conductivity, they have excellent electrochemical properties. Conducting polymer hydrogels have been widely used in MFCs and biosensors because of their excellent electrochemical activity and electrode properties [28–31]. The distinct properties of the hydrogel can be harnessed to provide the desired 3D architecture in an MFC anode [32].

In the current study, we used the hydrothermal method to fabricate polyaniline–sodium alginate–graphene oxide (PANI–SA–GO) composites directly coated on a carbon brush film (CB) without binders. Owing to the increase in the electrochemical reaction interface, PANI–SA–GO composite can effectively reduce the resistance of the solid–liquid interface and enhance the adhesion of microorganisms. Therefore, these composite materials give full play to the advantages of various components, thereby improving elec-

trochemical performance. This synthetic strategy will be instructive for the design of the composites, endowing them with the possibility for making sodium alginate/conducting polymer/carbon material composites for the bioanode of an MFC. The hydrogel bioanode was expected to improve the output and storage performance of the MFC. To the best of our knowledge, this study was the first attempt to develop a PANI–SA–GO/CB hydrogel electrode to enhance the current generation and storage of MFCs. Finally, MFCs capable of simultaneous power generation and energy storage may provide stable electric energy to low-power electrical equipment.

## 2. Experiment

### 2.1. Preparation of the PANI–SA–GO/CB Composite

Up to 0.03 g GO (99%; Zhongke Times Nanotechnology) was first dissolved in 6.8 mL of water and stirred for 30 min. Approximately 30 mg of p-phenylenediamine (PPD) was dissolved in 3 mL water. Then PPD solution was added to the aforementioned solution and stirred for 5 min. Subsequently, 0.74 mL of aniline was added to the aforementioned solution and stirred for 15 min. The solution was poured into the lining of a 50 mL hydrothermal reactor and placed at 0 °C in a water bath. Lastly, 0.003 g SA was dissolved in 8 mL of deionized water and then stirred to dissolve completely.

The treatment of carbon brush included the following steps. The carbon brush was soaked in acetone solution for 24 h, taken out of the solution, and rinsed with a lot of distilled water. Then the carbon brush was put in anhydrous ethanol and was subjected to ultrasonic cleaning twice, for 30 min each time. Finally, after rinsing with a lot of distilled water, it was placed in a drying oven with blast air at 60 °C for drying and reserve.

The treated CB was then immersed in a mixture of aniline and SA solution for 30 min. Exactly 1.902 g of ammonium persulfate was added to the SA solution, stirred for 5 min, placed at 0 °C in a water bath, quickly added into the aforementioned mixed solution, and was stirred for 30 s. After stirring, the mixed solution with SA was polymerized into the glue for 1 h without stirring. The whole reaction temperature was controlled at 0 °C. The polytetrafluoroethylene liner containing the hydrogel was then sealed in a stainless steel autoclave and placed in an oven at 180 °C for a 3 h reaction. When the reactor cooled naturally to room temperature, a cylindrical self-supporting hydrogel was obtained.

### 2.2. MFC Construction

The MFC construction and nutrient solution composition are presented in the Supporting Information. All MFC reactors were connected via external 1000 Ω resistors, running in a constant temperature oven at 30 °C. The total voltage and anode potential of each MFC were automatically recorded with a 20-channel USB data collector every 3 h. When the collected voltage data reached the peak value and gradually dropped to below 100 mV, the electrolyte of the two chambers was replaced until the MFC ran steadily. The reaction between the anode chamber and the cathode chamber is expressed as follows:

$$CH_3COO^- + 2H_2O \rightarrow 2CO_2 + 7H^+ + 8e^- \tag{1}$$

$$Fe(CN)_6^{3-} + e^- \rightarrow Fe(CN)_6^{4-} \tag{2}$$

### 2.3. Measurement and Characterization

The electrochemical performance of the MFC in a nutrient solution of a three electrode system was studied using electrochemical workstation (SP-240; Bio-Logic France): the prepared anode was used as the working electrode, graphite rods were used as counter electrodes, and the saturated Ag/AgCl (+197 mV, saturated with KCl, and corrected to a standard hydrogen electrode) electrode was the reference electrode. (The physical picture of the electrochemical testing process of MFC is shown in Figure S1) Cyclic voltammetry (CV) was tested in a nutrient solution with a sweeping speed of 1 mV/s, and the potential window determined by CV was −0.4–0.2 V. The polarization curve was collected by mea-

suring the cell voltage at different external resistances. The power (P) was calculated using P = IV. The polarization and power density curves were collected by changing the external resistance (R) from 100 $\Omega$ to 9000 $\Omega$. In the discharge experiment, the discharge voltage was $-0.1$ V, and the change of MFC current with time was recorded. The modified electrodes were characterized by field-emission scanning electron microscopy (SEM) (Sirion200, FEI Ltd., the Netherlands). Fourier transform infrared spectroscopy was used in this paper. Biomass on electrodes was quantified by measuring bacterial protein with the modified BCA method. In this paper, the PANI-SA-GO/CB hydrogel electrode was used as the modified anode, and a carbon felt electrode was used as the control electrode.

## 3. Results and Discussion

### 3.1. Physicochemical Characterization

Figure 1a,b present the multiple images of the CF film electrode. As shown in the figures, the surface of the carbon felt film contained numerous carbon fibers with considerably smooth surfaces. In Figure 1c,d, the PANI-SA-GO/CB hydrogel electrode clearly presents a three-dimensional network structure; the porosity and specific surface area were increased, rendering the electrode conducive to charge transfer and ion migration, with improved electrochemical performance. The large surface area of the electrode was beneficial for the microbial metabolism.

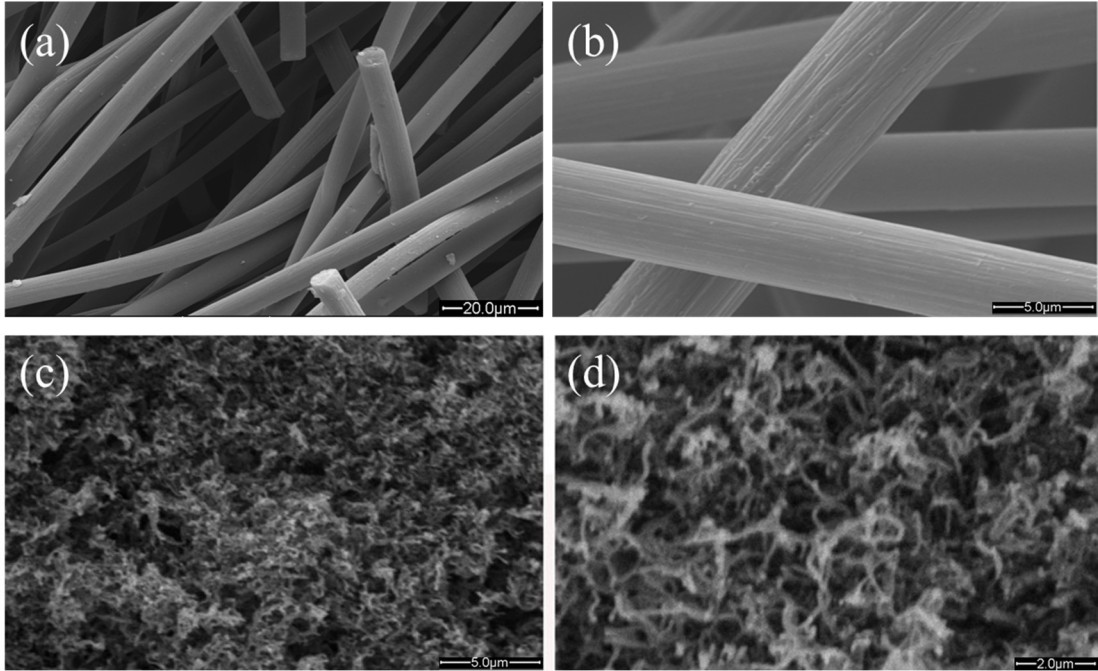

**Figure 1.** SEM images of the CF electrode ((**a**) 1000× and (**b**) 5000×) and the modified electrode ((**c**) 5000× and (**d**) 20,000×).

The infrared spectra of different modified materials is shown in Figure 2. Additionally, the intrinsic polyaniline was prepared in this research. In the PANI spectrum, the absorption peak at 1577 cm$^{-1}$ is the strong stretching vibration of the C=C bond on the quinone ring; the absorption peak at 1498 cm$^{-1}$ is the stretching vibration of the C=C bond on the benzene ring; the absorption peak at 1302 cm$^{-1}$ is the stretching vibration of the C-N bond; the absorption peak at 1138 cm$^{-1}$ is the in-plane bending vibration of the C-H bond on the benzene ring; and the absorption peak at 820 cm$^{-1}$ is the out-of-plane bending vibration of the C-H bond on the benzene ring [33–37]. In the SA spectrum, the absorption peak at 1608 cm$^{-1}$ is attributed to the asymmetric stretching vibration of –COO-, the absorption peak at 1419 cm$^{-1}$ is ascribed to the symmetric stretching vibration of -COO-, and the absorption peak at 1032 cm$^{-1}$ is C-O-C. In the GO spectrum, the stretching vibration

peak of C=O is observed at 1720 cm$^{-1}$, the bending vibration peak of C-OH occurs at 1621 cm$^{-1}$, and the vibration absorption peak of C-O-C appears at 1045 cm$^{-1}$. The PANI-SA-GO/CB spectrum basically contains the characteristic peaks of the substances, with a slight deviation of the peak positions.

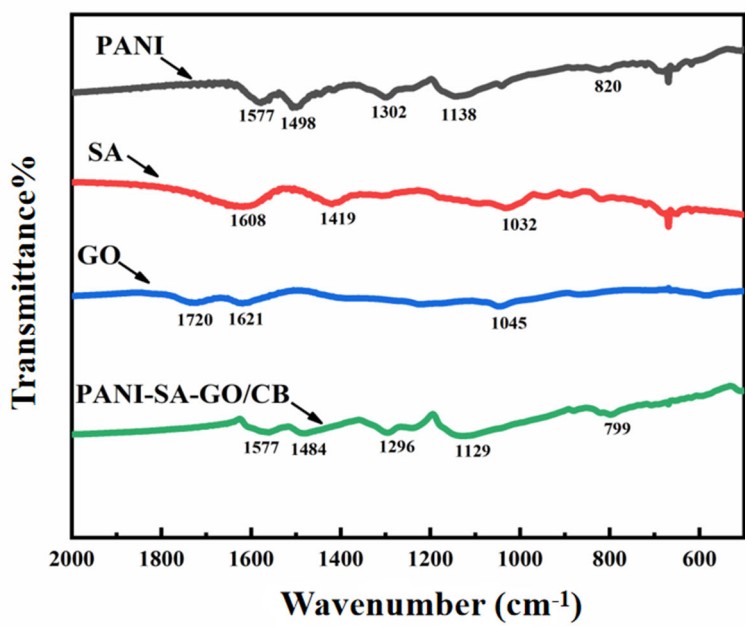

**Figure 2.** FTIR spectra of different modified materials.

To demonstrate the excellent performance of the MFC with a modified electrode, the MFC with a blank CF film anode was used as the control object. Figure 3 shows the output curve of MFCs with two anodes. The polarization curves of the modified anode and blank CF anode are shown in Figure 3a. The power density curves of the two anodes of MFCs are presented in Figure 3b. As shown in Figure 3a, the MFCs with the modified anode and that constructed with the blank CF anode showed open-circuit voltages of 724 and 644 mV, respectively. The open-circuit voltage represented the maximum voltage of the MFC can achieve. An MFC with high open-circuit voltage indicated good output voltage performance. When the output voltage of MFC reached 400 mV, the MFC with the PANI-SA-GO/CB anode showed a current density of 4.25 A/m$^2$. It was 3.08 times larger than that of the bare CF anode (1.38 A/m$^2$). The MFC with the modified anode showed a degree of polarization less than that of the blank CF anode in the same range of current density. During the current density ranging from 1 A/m$^2$ to 2 A/m$^2$, the MFCs equipped with the modified anode and bare CF anode had polarized voltages of 92 and 171 mV, indicating that the degree of MFC polarization of the modified anode was better than that of the bare CF anode. As shown in Figure 3b, the MFC's power density equipped with the two anodes initially rose and then decreased with the increasing current density. The MFC's power density of self-supporting hydrogel anode was much higher than that of the bare CF. The MFC's maximum power density with the modified anode reached 4970 mW/m$^3$, which was 2.72 times larger than that of the blank CF anode (1825 mW/m$^3$). A comparison between the PANI–SA–GO/CB composite anodes in the current study and those in previous results are shown in Table 1. When the MFC with a modified anode reached the maximum power density, the corresponding current density was 4.66 A/m$^2$. It was 1.79 times larger than that of the bare CF anode (2.61 A/m$^2$). The self-supported conductive PANI-SA-GO/CB hydrogel anode can provide a favorable metabolic environment for microorganisms, and the modified electrode can significantly improve the output power of the MFC.

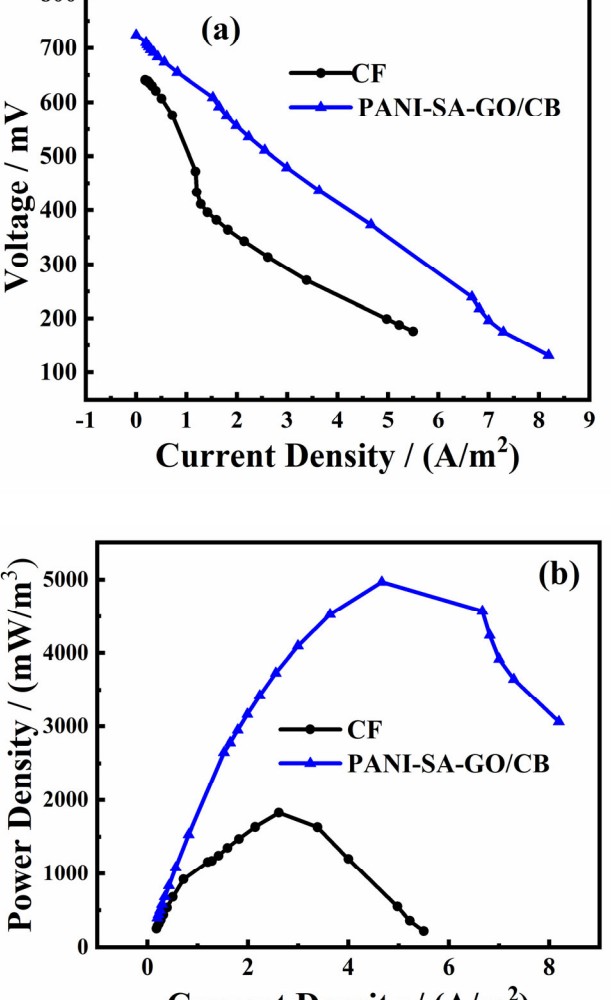

**Figure 3.** The output performance of two MFCs: (**a**) Polarization curves of MFCs with two anodes; (**b**) Power density curves of MFCs (the external resistance (R) from 100 Ω to 9000 Ω).

The excellent performance of the MFC with the modified anode was mainly due to the following: (1) Conductive polymer hydrogels have dual characteristics (advantages of hydrogels and organic conductors), and their inherent porous structure and high conductivity can provide guarantee for excellent electrode performance [29,38]. (2) SA is derived from a marine organism and has good biocompatibility and biodegradability. Direct electron transport through the outer membrane c-Cyts and indirect electron transport mediated by electron shuttles are the two main electron transport pathways of electrochemically active bacteria [39]. SA can promote electrons towards the active center of c-Cyts. The shortening of the distance between c-Cyts and electrons suggests that SA can accelerate electron transfer in microorganisms. The PANI-SA network structure helped to reduce the internal resistance, increase the electron transfer and electrolyte ion transport in the diffusion layer, and further improve the capacitance and power density of the MFCs. Large capacitance and energy density contributed to the output power density [40,41]. (3) The PANI-SA-GO/CB anode has a large specific surface area, which is conducive to microbial attachment and propagation. Therefore, the modification of the PANI-SA-GO anode can improve the anode surface area, facilitate the metabolism of microorganisms, and promote the increase of anode biomass.

**Table 1.** Comparison of the power density in both double-chamber (the schematic diagram of double-chamber MFC structure is shown in Supporting Information Figure S2) and single-chamber MFCs with different anode materials.

| Electrodes | Reactor Configuration | Power Density (mW/m$^2$) | References |
|---|---|---|---|
| PANI/WO$_3$/CF | Single | 980 | [42] |
| PANI/CNT/GF | Dual | 257 | [43] |
| PANI/3D-G | Dual | 768 | [27] |
| G/PANI/CC | Dual | 1390 | [9] |
| CPHs/CNTs | Dual | 1898 | [44] |
| G/PANI/Pt/CC | Dual | 2059 | [19] |
| PANI-SA-GO/CB | Dual | 4970 | This work |

### 3.2. Capacitive Behavior

Figure 4 presents the curve depicting the anode potential of the MFC equipped with two anodes with different charging durations. As shown in the figure, the charging durations of the curves from left to right are 5, 30, and 60 min. As the charge time increases, both anode potentials acquire increasingly negative values. However, the anode potential of the blank carbon felt film decreases faster with an increase in charging time. During the 60 min charging period, the anode potential of the CF reached its lowest value (−0.45 V). The PANI-SA-GO/CB hydrogel anode showed a slower reduction in potential. The anode potential only reached −0.39 V, during the charging time of 60 min, indicating that the modified anode exhibited a stronger effect as a capacitive MFC anode and could store more charge. The charge generated by the microorganism could initially be stored in the capacitive anode, hence the slow decrease in the potential of the composite anode.

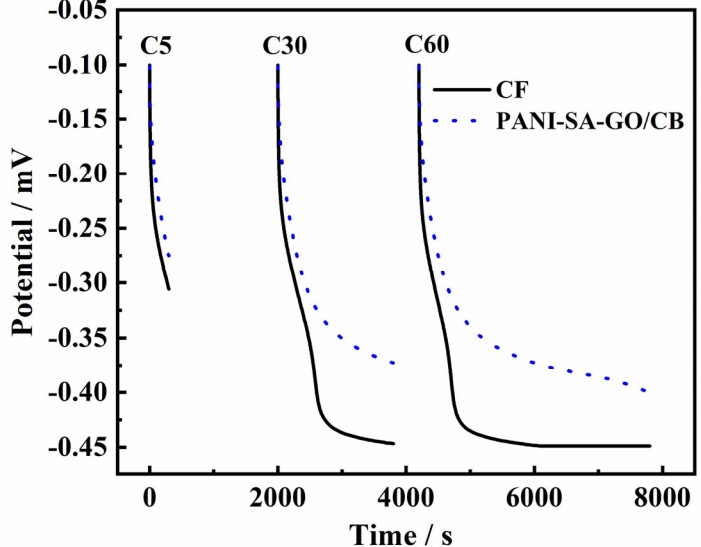

**Figure 4.** Time–potential tests of MFCs with two anodes. (The MFCs were under open circuit condition).

Figure 5a–c show the discharge curves of two anodes at the control potential −0.1 V. As is shown, all the discharge curves showed a peak current ($i_h$) at the beginning of the discharge test, followed by a rapid current decay until it approached a relatively constant value ($i_s$). The peak current density was highly dependent on the specific capacitance.

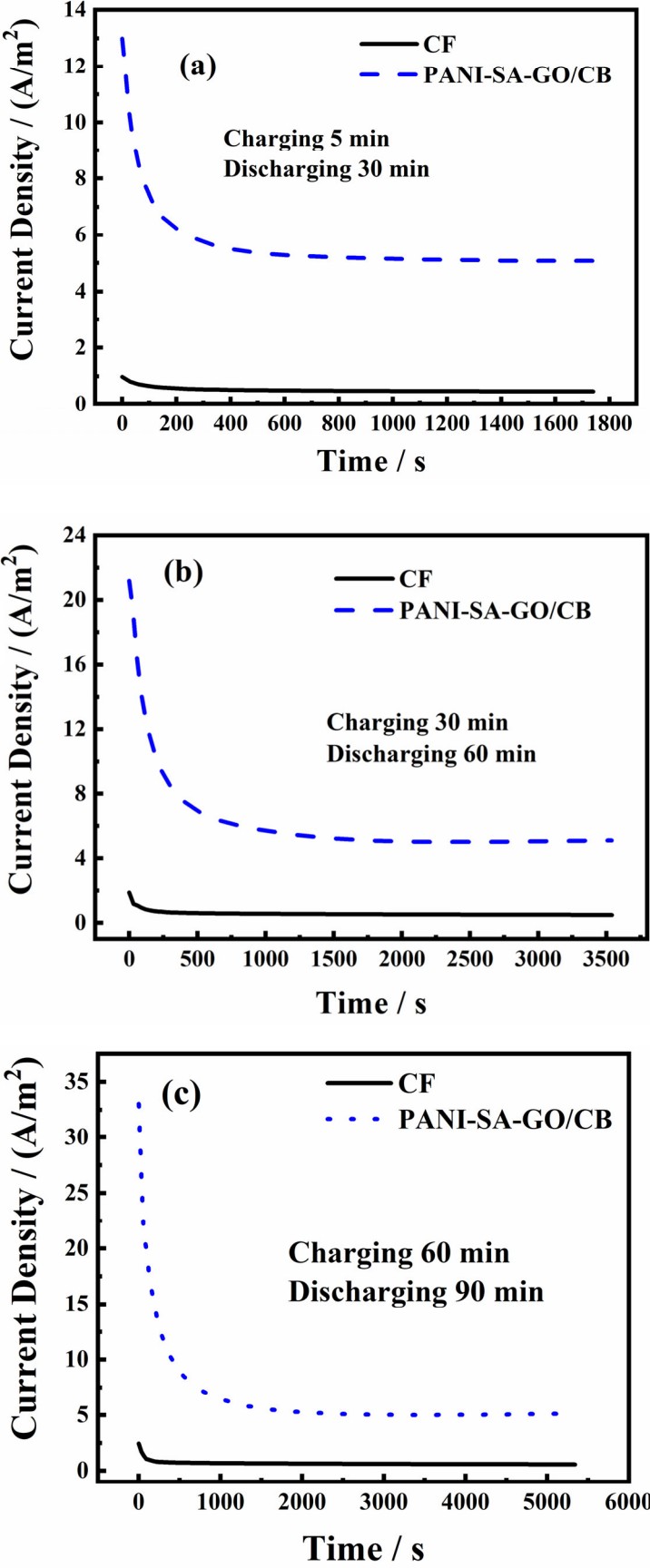

**Figure 5.** Discharge curves (**a**–**c**) tested at −0.1 V for two anodes. (The MFCs were under the closed-circuit condition).

As shown in Figure 5a–c, both the initial current density and the stationary current density of the modified anode were higher than that of the CF anode. Figure 5 also shows that as the charge and discharge periods were extended, the amount of electric charge stored by the anode was increased, and the amount of electric charge stored by the modified hydrogel anode was larger than that of the bare CF anode. These findings indicated that the modified composite material played an important role in cultivating microorganisms and could be used as a suitable anode material to further enhance the energy storage performance of MFCs. The modified hydrogel anode could greatly improve the storage capacity of electric energy. The specific values of stored charge, peak current density, and stationary current density are listed in Table 2.

**Table 2.** The parameter of discharging curves with blank anode and modified anode.

| Electrodes | Parameters | C5 min/D30 min | C30 min/D60 min | C60 min/D90 min |
|---|---|---|---|---|
| CF | $i_h$ (A/m$^2$) | 0.98 | 2.29 | 2.45 |
| | $i_s$ (A/m$^2$) | 0.46 | 0.55 | 0.58 |
| | $Q_s$ (C/m$^2$) | 87.07 | 308.94 | 423.05 |
| | $Q_m$ (C/m$^2$) | 887.47 | 2255.94 | 3520.25 |
| PANI-SA-GO/CB | $i_h$ (A/m$^2$) | 12.99 | 21.18 | 33.03 |
| | $i_s$ (A/m$^2$) | 5.07 | 5.11 | 5.12 |
| | $Q_s$ (C/m$^2$) | 868.56 | 3397.96 | 6378.41 |
| | $Q_m$ (C/m$^2$) | 9694.53 | 21,487.71 | 33,756.32 |

As shown in Table 2, the MFC with the modified anode showed a significant improvement in electrical storage performance. When the charge–discharge period was set to 5–30 min, the total charge of the modified anode was 9694.53 C/m$^2$. It was 10.9 times higher than that of the CF anode (887.47 C/m$^2$). At a charging–discharging time of 60–90 min, the highest stored charge of the modified anode (6378.41 C/m$^2$) was 15.08 times larger than that of the CF (423.05 C/m$^2$). The modified capacitive anode had a peak current density of 33.03 A/m$^2$. It was 13.48 times higher than that of the CF anode (2.45 A/m$^2$). Bioanodes modified with capacitive materials exhibit capacitive behavior comparable to that of biocapacitors. When the circuit is open, the microbes attached to the anode surface degrade the organic matter to produce an electric charge, which is first stored in the capacitor material on the anode surface. When electricity is needed, both the stored charge and the electrons produced can be released quickly. It can not only overcome the mismatch between electricity production and demand, but also provide large current and high-power output in a short time [45–48]. These results indicate that the modified composite provided superior capacitance performance to that of the CF anode. Moreover, the modified anode could facilitate the transmission of electrons from microorganisms to the anode surface, as well as promote adhesion, growth, and reproduction of microorganisms on the anode surface. Consequently, the discharge and energy storage of the MFC are improved.

Figure 6 shows the comparison of protein contents of two bioanodes. The figure shows the PANI-SA-GO hydrogel anode and blank carbon felt (CF) anode. As can be seen from the figure, the protein content of the PANI-SA-GO/CB anode was 70.24 mg/cm$^3$, which was 7.47 times of that of the blank CB anode, compared at 9.4 mg/cm$^3$. This was because PANI material and SA material had good biocompatibility, which was conducive to microbial attachment. The modified anode prepared had a 3D porous network structure and high specific surface area, which could provide more space for microorganisms to attach.

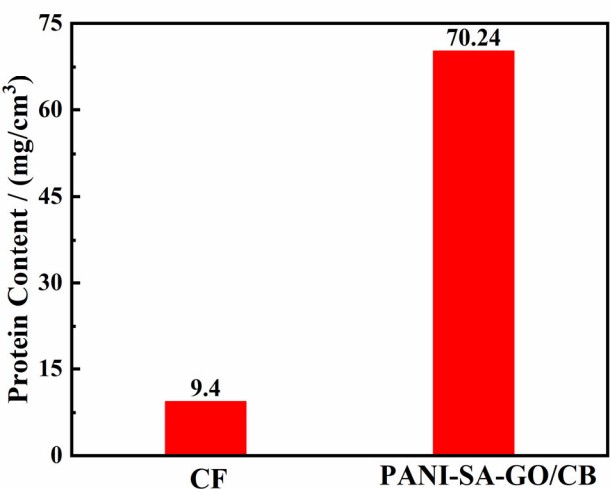

**Figure 6.** Protein content of MFC with two anodes.

## 4. Conclusions

In this study, we designed a PANI-SA-GO/CB hydrogel as an efficient MFC anode material capable of generating and storing energy at the same time. Additionally, the advantages of hydrogels and organic conductors are complemented by the combination of polysaccharide polymer SA and graphene oxide material to prepare a three-dimensional nano hydrogel composite anode with a self-supporting structure. In addition to its compactness, the structure was helpful in increasing the specific surface area of the anode, and it promoted the electrogenesis bacteria adhesion and metabolism. Both the energy storage and the maximum power density of the MFC equipped with the PANI-SA-GO/CB hydrogel increased substantially, compared with the control. The MFC equipped with the PANI-SA-GO/CB anode was 4970 mW/m$^3$, which was 2.72 times higher than that of the blank CF anode (1825 mW/m$^3$). Moreover, the biomass of the modified anode was 7.47 times higher than that of CF. These results proved that the PANI-SA-GO/CB hydrogel composite provided an effective approach to enhancing power production and energy storage in MFCs.

**Supplementary Materials:** The following supporting information can be downloaded at: https://www.mdpi.com/article/10.3390/coatings13040790/s1, Figure S1: The physical picture of the electrochemical testing process of MFC; Figure S2: The schematic diagram of double-chamber MFC structure.

**Author Contributions:** Y.W. (Writing—review and editing): Preparation, creation and/or presentation of the published work by those from the original research group, specifically critical review, commentary, and revision—including pre- and post-publication stages. H.Y. (Writing—review and editing): Preparation, creation and/or presentation of the published work, specifically writing the initial draft (including substantive translation). J.W. (Formal analysis): Application of statistical, mathematical, computational, and other formal techniques to analyze and synthesize study data. J.D. (Conceptualization of ideas): Formulation and evolution of overarching research goals and aims. Y.D. (Resources): Provision of study materials, reagents, materials, patients, laboratory samples, animals, instrumentation, computing resources, and other analysis tools. All authors have read and agreed to the published version of the manuscript.

**Funding:** The project was supported by the Innovation and Entrepreneurship Training Program for College Students of Harbin University of Commerce, No. 202110240048; Heilongjiang Natural Science Foundation joint guide project No. LH2020E027.

**Institutional Review Board Statement:** Not applicable.

**Informed Consent Statement:** Not applicable.

**Data Availability Statement:** The data presented in this study are available in this article.

**Acknowledgments:** Thanks to Ye Tian's project fund for supporting this article (Heilongjiang Natural Science Foundation joint guide project No. LH2020E027).

**Conflicts of Interest:** The authors declare no conflict of interest.

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
