# Peer review of "Self-Supporting Conductive Polyaniline–Sodium Alginate–Graphene Oxide/Carbon Brush Hydrogel as Anode Material for Enhanced Energy in Microbial Fuel Cells"

_coatings, doi:10.3390/coatings13040790_

Round 1

Reviewer 1 Report

The authors present the work entitled "Self-supporting conductive polyanilinesodium alginategraphene oxide/carbon brush hydrogel as anode material for enhanced energy in microbial fuel cells", for being considered in coatings. After revising this work, I have some concerns that makes me to recommend rejection.
Herein my comments:
1. In experimental, authors said that they employed a treated carbon brush but they do not give information about the treatment.

2. In electrochemical characterization, please correct “chemical workstation” to “electrochemical workstation”

3. Discharge experiments are usually performed demanding different current values to the cell/battery, and the potential required to achieve that demand is reported in form of a discharge curve. The authors said that they employed -0.1 V, I am wondering if the test was performed in galvanostatic mode of the workstation, or it is truly a polarization curve rather than a discharge curve. Please explain.

4. Line 162. “electrode clearly presents a three-dimensional network structure; the porosity and specific surface area are increased, rendering the electrode conducive”. How the electric conductivity is improved with these morphological changes?

5. Correct “wavement” to “wavenumber” in Figure 2.

6. FTIR did not give information about the kind of PANI which is obtained (hemeraldine, etc.) Please consider to add Raman spectroscopy.

7. I made some multiplications of data presented in Fig. 3a to obtain power densities, and I couldn’t get the results presented in Fig. 3b, please revise. Additionally, how the power density was normalized to m3?

8. line 248: I do not how the observation of a reduction peak is an indicative of a better catalytic activity.

9. My main concern is with the novelty of the work. The corresponding author (Y. Wang) has similar works like Energy 202 (2020) 117780, and they are not considered in the introduction section. Moreover, my most serious concern is with some plots, particularly Fig. 4a, blue line for PANI-SA-GO/CB which is identical to that presented in the above mentioned paper in Fig. 5A as “PANI-SA/CB”.

Author Response

Thank you very much for the Reviewers’ comments concerning our manuscript, these comments are all valuable and very helpful for revising and improving our paper. We upload a separate file“Response to Reviewers”. 

Reviewer 2 Report

Dear Authors

After reviewing your manuscript entitled ‘Self-supporting conductive polyaniline–sodium alginate–graphene oxide/carbon brush hydrogel as anode material for enhanced energy in microbial fuel cells’ I would like to give you a few remarks that should be taken under consideration.

Remarks

1.     I would like to ask Authors to decide if they use the present or past tense in the text

2.     Authors should remember that a Figure or Table should be cited in the text before it appears (see Fig.3, 4, 5, 6).

3.     Page 3, line 99 – change ‘OWING’ to ‘Owing’

4.     Page 5, line 169, Figure 2 – description of the y-axis is missing; change ‘wavement’ to ‘wavenumber’; improve the quality of the presented spectra, in their current form they are too flat for the reader to clearly see the marked peaks

5.     Page 11, Figure 7 - There is no citation of Figure 7 in the text – please complete it.

6.     Page 11, line 322 – change ‘conclusion’ to ‘Conclusion’

7.     Page 11-12, ‘Conclusion’ section – the authors in the introduction indicate that the anode they developed is new and has not been described in the literature so far. In that case, it is an achievement that should be highlighted in this section.

8.     Page 12, lines 328-329 – ‘Moreover, the biomass of the PANI–SA–GO/CB hydrogel was 7.47 times that of CF’ – in my opinion, one word is missing in this underline statement ‘higher or lower’ – please complete

9.     Page 12  - the subsection of the Authors contribution is missing – please provide the type of participation of each of the co-authors in the creation of the work

For details see the Coatings website (https://www.mdpi.com/journal/coatings/instructions)

For research articles with several authors, a short paragraph specifying their individual contributions must be provided. The following statements should be used "Conceptualization, X.X. and Y.Y.; Methodology, X.X.; Software, X.X.; Validation, X.X., Y.Y. and Z.Z.; Formal Analysis, X.X.; Investigation, X.X.; Resources, X.X.; Data Curation, X.X.; Writing – Original Draft Preparation, X.X.; Writing – Review & Editing, X.X.; Visualization, X.X.; Supervision, X.X.; Project Administration, X.X.; Funding Acquisition, Y.Y.’

10.  Page 12 – 14 – ‘References’  - the newest paper listed in this section became from 2020. Please update the references with the newest literature reports (from the 2022 and 2023).

Author Response

(The authors gave the same response as above.)

Reviewer 3 Report

The paper is interesting to read and is quite in the perspective of bio sources of energy and waste use to produce energy, so very mach in line of current topic.

It would be nice giving the reader a schematic in the introduction of the actual structure of the MFC and comparison to previous constructions as in the final part of the paper the results in terms of energy production are compared so it would be interesting to have a visual about the idea how authors thought about it.

In paragraph 2.3 authors discuss the experimental setup and I feel that an schematic representation of the setup would be very well come .

In Figure 1 the bar is not very visible, it would be probably a good idea to make it more present in the image.

In Figure 1 authors say that FTIR corresponds to different electrodes, I understood that all those elements form one composite having specific electrical properties. It would be good to clarify that.

Table 1 could correlate to the remark at the beginning of the text, and it would be useful giving those schematic representations either at the introduction or even more evident here in the comparison of results.

In Figure 4 only 2 anodes are compared the carbon and the actual anode, so the text under the image should correspond to this. Different anodes means that there are several of them.

Regarding the results it would be interesting to comment on the stability of the system as it consists of living organisms and how they support the electric field, does this service kill some of them and if this conditions are viable for long term use.

I was happy with results and I feel that some clarifications should be good for the better understanding of the paper.

Author Response

(The authors gave the same response as above.)

Reviewer 4 Report

Dear Authors,

The Authors show a PANI–SA–GO/CB hydrogel as an effective MFC anode material capable of simultaneous electricity generation and energy storage. In addition to its compactness, the structure increased the specific surface area of the anode and promoted microbial adhesion and growth. Both the energy storage and the maximum power density of the MFC equipped with the PANI–SA–GO/CB hydrogel increased substantially, compared with the control.

The description of the work is acceptable. Overall impression is that this manuscript can be recommended for publication after MAJOR revision in Coatings especially considering the scope and topics of this journal. However, I would like to point out to several details:

  1. It is not clear, i.e. you should emphasize what is novelty in your paper worth to publish? Correct this.
  2. Captions for figures need to be more detailed and consist of some experimental conditions. Correct this.
  3. In the conclusions, in addition to summarizing the actions taken and results, please strengthen the explanation of their significance. It is recommended to use quantitative reasoning comparing with appropriate benchmarks, especially those stemming from previous work. Correct this.
  4. Do the authors have the figure of the reproducibility? It is very important for system like this. Correct this.
  5. English language should be corrected by a professional lector. A proof reading by a native English speaker should be conducted to improve both language and organization quality.

I hope that I helped in making this manuscript worth of publishing. I wish a lot of success to the authors in making this manuscript much better.

With kind regards!

Reviewer

Author Response

(The authors gave the same response as above.)

Round 2

Reviewer 4 Report

DearAuthors,

Interesting results are well presented. The description of the work is acceptable. The length of the manuscript is appropriate. Discussion and conclusion is detailed. In my opinion this manuscript can be PUBLISH in Coatings especially considering the scope and topics of this journal. The authors correct all suggestions that reviewers gave about article.

I wish a lot of success to the authors.

Regards!

Reviewer